# Dynamic Sasvi: Strong Safe Screening for Norm-Regularized Least Squares

**Hiroaki Yamada**
Kyoto University
hyamada@ml.ist.i.kyoto-u.ac.jp

**Makoto Yamada**
Kyoto University and RIKEN AIP
myamada@i.kyoto-u.ac.jp

## Abstract

A recently introduced technique, called "safe screening," for a sparse optimization problem allows us to identify irrelevant variables in the early stages of optimization. In this paper, we first propose a flexible framework for safe screening based on the Fenchel–Rockafellar duality and then derive a strong safe screening rule for norm-regularized least squares using the proposed framework. We refer to the proposed screening rule for norm-regularized least squares as "dynamic Sasvi" because it can be interpreted as a generalization of Sasvi. Unlike the original Sasvi, it does not require the exact solution of a more strongly regularized problem; hence, it works safely in practice. We show that our screening rule always eliminates more features compared with the existing state-of-the-art methods.

## 1 Introduction

Sparse models such as Lasso [23] and group Lasso [26] have been widely studied in the areas of statistics and machine learning and used for various applications such as compressed sensing [6] and biomarker discovery [4]. Although sparse models can be formulated as a simple convex optimization problem, the computational cost can be large if the numbers of samples and dimensions are extremely large.

To address this problem, a technique called safe screening has been introduced [10] for Lasso problems. Specifically, it eliminates variables that are guaranteed to be zero in the Lasso solution before solving the original Lasso optimization problem. Many safe screening methods have been proposed for various problems [10, 19, 24, 14, 25]. These are called sequential screening rules because they require the solution to a more strongly regularized problem. A recent technique, called dynamic screening, has been proposed to eliminate variables through an estimated solution in an iterative solver [3]. In particular, Gap Safe [7, 16], a dynamic screening framework, is widely used owing to its generality and efficiency [17, 21, 1, 20, 18].

In this paper, we propose a dynamic safe screening algorithm that is stronger than Gap Safe for the *Lasso-Like* problem, which includes norm-regularized least squares. To this end, we first propose a general screening framework based on the Fenchel–Rockafellar duality and then derive *Dynamic Sasvi*, a strong safe screening rule for *Lasso-like* problems. Our framework can be regarded as a generalization of the Gap Safe framework, and thus, we can derive Gap Safe simply using our results. Moreover, owing to this generalization, we can use the strong problem adaptive inequality. The derived screening rule for *Lasso-like* problems can be seen as a dynamic variant of safe screening with variational inequalities (Sasvi) [14], which is a sequential screening rule for Lasso. Therefore, we refer to this as Dynamic Sasvi. Unlike the original Sasvi, dynamic Sasvi does not require an exact solution to the problem with another hyper-parameter and, hence, operates safely in practice. Moreover, we propose the use of dynamic enhanced dual polytope projection (EDPP) [24], which is a relaxation of dynamic Sasvi due to the introduction of a minimum radius sphere. We show, both

35th Conference on Neural Information Processing Systems (NeurIPS 2021).

theoretically and experimentally, that the screening power and computational costs of Dynamic Sasvi and Dynamic EDPP compare favorably with those of other state-of-the-art Gap Safe methods.

**Contribution:** The contributions of our study are summarized as follows.

- We propose a flexible screening framework based on the Fenchel–Rockafellar duality, which is a generalization of the Gap Safe framework [17].
- We propose two novel dynamic screening rules for norm-regularized least squares: a dynamic variant of Sasvi [14] and a dynamic variant of EDPP.
- We show that Dynamic Sasvi always eliminates more features and increases the speed of the solver compared with Gap Safe [7, 17] .

## 2    Preliminary

In this section, we first formulate the problem and introduce the key techniques used in this study.

### 2.1    Notation

Given $h : \mathbb{R}^m \to [-\infty, \infty]$, the domain of $h$ is defined by $\mathrm{dom}(h) := \{\boldsymbol{z} \in \mathbb{R}^m \mid |h(\boldsymbol{z})| < \infty\}$ and $h^\star : \mathbb{R}^m \to [-\infty, \infty]$, which is the Fenchel conjugate of $h$, is defined as $h^\star(\boldsymbol{v}) := \sup_{\boldsymbol{z} \in \mathbb{R}^m} \boldsymbol{v}^\top \boldsymbol{z} - h(\boldsymbol{z})$. If $h$ is proper, the Fenchel–Young inequality,

$$h(\boldsymbol{z}) + h^\star(\boldsymbol{v}) \geq \boldsymbol{v}^\top \boldsymbol{z}, \tag{1}$$

can be proven directly based on definition of the Fenchel conjugate. The subdifferential of a proper function, $h : \mathbb{R}^m \to (-\infty, \infty]$, at $\boldsymbol{z}$ is given as

$$\partial h(\boldsymbol{z}) := \{\boldsymbol{v} \in \mathbb{R}^m \mid \forall \boldsymbol{w} \in \mathbb{R}^m \; \boldsymbol{v}^\top(\boldsymbol{w} - \boldsymbol{z}) + h(\boldsymbol{z}) \leq h(\boldsymbol{w})\}.$$

The next proposition is important for deriving safe-screening algorithms.

**Proposition 1** *Assume that $h : \mathbb{R}^m \to (-\infty, \infty]$ is a proper lower semi-continuous convex function and $\boldsymbol{z}, \boldsymbol{v} \in \mathbb{R}^m$. We then have*

$$\boldsymbol{v} \in \partial h(\boldsymbol{z}) \iff h(\boldsymbol{z}) + h^\star(\boldsymbol{v}) = \boldsymbol{v}^\top \boldsymbol{z} \iff \boldsymbol{z} \in \partial h^\star(\boldsymbol{v}).$$

See Section 16 of [2] for the proof. For convex set $C \subset \mathbb{R}^m$, the relative interior of $C$ is defined as

$$\mathrm{relint}(C) := \{v \in C \mid \forall w \in C \; \exists \epsilon > 0 \text{ s.t. } v + \epsilon(v - w) \in C\}.$$

### 2.2    Problem Formulation

In this study, we consider an optimization problem, formulated as

$$\underset{\boldsymbol{\beta} \in \mathbb{R}^d}{\mathrm{minimize}} \; f(\boldsymbol{X}\boldsymbol{\beta}) + g(\boldsymbol{\beta}), \tag{2}$$

where $\boldsymbol{\beta} \in \mathbb{R}^d$ is the optimization variable, $\boldsymbol{X} \in \mathbb{R}^{n \times d}$ is a constant matrix, and $f : \mathbb{R}^n \to (-\infty, \infty]$ and $g : \mathbb{R}^d \to (-\infty, \infty]$ are proper lower semi-continuous convex functions. We assume

$$\exists \boldsymbol{\beta} \in \mathrm{relint}(\mathrm{dom}(g)) \text{ s.t. } \boldsymbol{X}\boldsymbol{\beta} \in \mathrm{relint}(\mathrm{dom}(f))$$

and the existence of the optimal point, i.e.,

$$\exists \hat{\boldsymbol{\beta}} \in \mathrm{dom}(P) \text{ s.t. } P(\hat{\boldsymbol{\beta}}) = \inf_{\boldsymbol{\beta} \in \mathbb{R}^d} P(\boldsymbol{\beta}),$$

where $P : \mathbb{R}^d \to \mathbb{R}$ is defined as $P(\boldsymbol{\beta}) = f(\boldsymbol{X}\boldsymbol{\beta}) + g(\boldsymbol{\beta})$. Moreover, we focus on the cases in which $g$ induces sparsity. Although all theorems in this paper hold, we cannot eliminate any variables without sparsity.

This is a popular class of optimization problem, with the most popular example being Lasso [23]:

$$\underset{\boldsymbol{\beta} \in \mathbb{R}^d}{\mathrm{minimize}} \; \frac{1}{2} \|\boldsymbol{y} - \boldsymbol{X}\boldsymbol{\beta}\|_2^2 + \lambda \|\boldsymbol{\beta}\|_1.$$

Many extensions of Lasso, including Group-Lasso [26], elastic net [27], and sparse logistic regression [15], are in this class. The dual problems of a support vector machine [5] and a support vector regression [22] are also in this class.

## 2.3 Dual Problem

In the derivation of a safe screening rule for the optimization problem, Eq. (2), the Fenchel–Rockafellar dual formulation, plays an important role.

**Theorem 2** *(Fenchel–Rockafellar Duality) If all assumptions for the optimization problem* (2) *are satisfied, we have the following:*

$$\min_{\boldsymbol{\beta} \in \mathbb{R}^d} f(\boldsymbol{X}\boldsymbol{\beta}) + g(\boldsymbol{\beta}) = \max_{\boldsymbol{\theta} \in \mathbb{R}^n} -f^\star(-\boldsymbol{\theta}) - g^\star(\boldsymbol{X}^\top \boldsymbol{\theta}). \tag{3}$$

The proof of Theorem 2 is provided in the appendix. We denote $-f^\star(-\boldsymbol{\theta}) - g^\star(\boldsymbol{X}^\top \boldsymbol{\theta})$ by $D(\boldsymbol{\theta})$. For primal/dual solutions, there are many conditions that are equivalent to optimality. Herein, we provide a list of such conditions for convenience.

**Proposition 3** *(Optimal Condition) If all assumptions for the optimization problem* (2) *are satisfied, the following are equivalent.*

*(a)* $\hat{\boldsymbol{\beta}} \in \operatorname{argmin}_{\boldsymbol{\beta} \in \mathbb{R}^d} P(\boldsymbol{\beta}) \wedge \hat{\boldsymbol{\theta}} \in \operatorname{argmax}_{\boldsymbol{\theta} \in \mathbb{R}^n} D(\boldsymbol{\theta})$

*(b)* $P(\hat{\boldsymbol{\beta}}) = D(\hat{\boldsymbol{\theta}})$

*(c)* $f(\boldsymbol{X}\hat{\boldsymbol{\beta}}) + f^\star(-\hat{\boldsymbol{\theta}}) = -\hat{\boldsymbol{\theta}}^\top \boldsymbol{X}\hat{\boldsymbol{\beta}} = -g(\hat{\boldsymbol{\beta}}) - g^\star(\boldsymbol{X}^\top \hat{\boldsymbol{\theta}})$

*(d)* $-\hat{\boldsymbol{\theta}} \in \partial f(\boldsymbol{X}\hat{\boldsymbol{\beta}}) \wedge \boldsymbol{X}^\top \hat{\boldsymbol{\theta}} \in \partial g(\hat{\boldsymbol{\beta}})$

*(e)* $\boldsymbol{X}\hat{\boldsymbol{\beta}} \in \partial f^\star(-\hat{\boldsymbol{\theta}}) \wedge \hat{\boldsymbol{\beta}} \in \partial g^\star(\boldsymbol{X}^\top \hat{\boldsymbol{\theta}})$

**(Proof)** (a) $\iff$ (b) is directly derived from the strong duality. (b) $\iff$ (c) is derived from the Fenchel–Young inequality (1). (c) $\iff$ (d) $\iff$ (e) are derived from Proposition 1. $\qquad \square$

## 2.4 Relationship of Dual Safe Region and Screening

For the optimization problem (2), we can eliminate some features by constructing a simple region that contains $\hat{\boldsymbol{\theta}}$. Assume that the dual optimal point, $\hat{\boldsymbol{\theta}}$, is within region $\mathcal{R}$. According to Proposition 3, we have

$$\hat{\boldsymbol{\beta}} \in \partial g^\star(\boldsymbol{X}^\top \hat{\boldsymbol{\theta}}) \subset \bigcup_{\boldsymbol{\theta} \in \mathcal{R}} \partial g^\star(\boldsymbol{X}^\top \boldsymbol{\theta}).$$

Hence, if $\bigcup_{\boldsymbol{\theta} \in \mathcal{R}} \partial g^\star(\boldsymbol{X}^\top \boldsymbol{\theta}) \subset \{\boldsymbol{\beta} \mid \boldsymbol{\beta}_i = 0\}$ holds, we obtain $\hat{\boldsymbol{\beta}}_i = 0$. A simple example is the following corollary.

**Example 4** *Consider an optimization problem, i.e., Eq.* (2) *with* $g(\boldsymbol{\beta}) = \|\boldsymbol{\beta}\|_1$. *Assume that* $\hat{\boldsymbol{\theta}} \in \mathcal{R}$. *Then, we have*

$$\max_{\boldsymbol{\theta} \in \mathcal{R}} |\boldsymbol{x}_i^\top \boldsymbol{\theta}| < 1 \implies \hat{\boldsymbol{\beta}}_i = 0.$$

**(Proof)** Based on the definition of $g$, we have $\partial g^\star(\boldsymbol{X}^\top \boldsymbol{\theta}) \subset \{\boldsymbol{\beta} \mid \boldsymbol{\beta}_i = 0\} \iff |\boldsymbol{x}_i^\top \boldsymbol{\theta}| < 1$. When $\max_{\boldsymbol{\theta} \in \mathcal{R}} |\boldsymbol{x}_i^\top \boldsymbol{\theta}| < 1$, we have $\hat{\boldsymbol{\beta}} \in \bigcup_{\boldsymbol{\theta} \in \mathcal{R}} \partial g^\star(\boldsymbol{X}^\top \boldsymbol{\theta}) \subset \{\boldsymbol{\beta} \mid \boldsymbol{\beta}_i = 0\}$. $\qquad \square$

To eliminate features safely, we should first construct a simple region $\mathcal{R} \subset \mathbb{R}^n$ which contains $\hat{\boldsymbol{\theta}}$ and then determine whether $\bigcup_{\boldsymbol{\theta} \in \mathcal{R}} \partial g^\star(\boldsymbol{X}^\top \boldsymbol{\theta}) \subset \{\boldsymbol{\beta} \in \mathbb{R}^d \mid \boldsymbol{\beta}_i = 0\}$ is guaranteed. In this study, we provide a novel general framework for constructing a simple safe region, $\mathcal{R} \subset \mathbb{R}^n$. By combining it with existing methods to calculate an upper bound of $\bigcup_{\boldsymbol{\theta} \in \mathcal{R}} \partial g^\star(\boldsymbol{X}^\top \boldsymbol{\theta})$ (cf. [17] for the un-overlapping group L1 norm and sparse group L1 norm, [1] for the ordered weighted L1 norm), we can formulate strong safe screening rules.

# 3 General Framework for Constructing Safe Region

Herein, we propose a general framework for constructing a dual region that contains the solution to the optimization problem expressed by Eq. (3). Our framework consists of a general lower bound and a problem-adaptive upper bound of the optimal value. Hence, we can derive a narrower region than the framework with a general upper bound under certain situations.

The general lower-bound is derived from the optimal condition and the $L$-strong convexity. Assume that $f^\star$ is $L$-strongly convex ($L \geq 0$). Then, as $D$ is $L$-strongly concave and $\hat{\boldsymbol{\theta}}$ is the optimal point, we have $D(\tilde{\boldsymbol{\theta}}) \leq D(\hat{\boldsymbol{\theta}}) - \frac{L}{2}\|\hat{\boldsymbol{\theta}} - \tilde{\boldsymbol{\theta}}\|_2^2$ for $\forall \tilde{\boldsymbol{\theta}} \in \mathbb{R}^n$. Thus, we have

$$\hat{\boldsymbol{\theta}} \in \{\boldsymbol{\theta} \mid \frac{L}{2}\|\boldsymbol{\theta} - \tilde{\boldsymbol{\theta}}\|_2^2 + D(\tilde{\boldsymbol{\theta}}) \leq D(\boldsymbol{\theta})\}.$$

Because this region is too complicated for screening, we use a simple upper bound of $D(\boldsymbol{\theta})$ to construct a simple safe region.

**Theorem 5** *Consider the optimization problem expressed by Eq. (3) and assume that $f^\star$ is $L$-strongly convex ($L \geq 0$). Let $\hat{\boldsymbol{\theta}}$ be the solution to Eq. (3). Assume that $D(\boldsymbol{\theta})$ is upper-bounded by $u(\boldsymbol{\theta})$, i.e., $\forall \boldsymbol{\theta} \in \mathbb{R}^n \ D(\boldsymbol{\theta}) \leq u(\boldsymbol{\theta})$. Then, for $\forall \tilde{\boldsymbol{\theta}} \in \mathbb{R}^n$, we have*

$$\hat{\boldsymbol{\theta}} \in \mathcal{R}(\tilde{\boldsymbol{\theta}}, u) = \{\boldsymbol{\theta} \mid \frac{L}{2}\|\boldsymbol{\theta} - \tilde{\boldsymbol{\theta}}\|_2^2 + D(\tilde{\boldsymbol{\theta}}) \leq u(\boldsymbol{\theta})\}.$$

The complexity of $\mathcal{R}(\tilde{\boldsymbol{\theta}}, u)$ depends on the complexity of $u$. For example, if $u$ is linear, then $\mathcal{R}(\tilde{\boldsymbol{\theta}}, u)$ is a sphere. We can construct a narrow, simple, and safe region with a tight simple upper-bound $u$. In fact, the Gap Safe Sphere region [7, 17] can be derived easily from this theorem and weak duality.

**Corollary 6** *(Gap Safe Sphere) Consider the optimization problem described by Eq. (3) and assume that $f^\star$ is $L$-strongly convex ($L \geq 0$). Let $\hat{\boldsymbol{\theta}}$ be the solution to Eq. (3). For $\forall \tilde{\boldsymbol{\beta}} \in \mathbb{R}^d$ and $\forall \tilde{\boldsymbol{\theta}} \in \mathbb{R}^n$, we have*

$$\hat{\boldsymbol{\theta}} \in \{\boldsymbol{\theta} \mid \frac{L}{2}\|\boldsymbol{\theta} - \tilde{\boldsymbol{\theta}}\|_2^2 + D(\tilde{\boldsymbol{\theta}}) \leq P(\tilde{\boldsymbol{\beta}})\}.$$

**(Proof)** Based on a weak duality, we have $\forall \boldsymbol{\theta} \ D(\boldsymbol{\theta}) \leq P(\tilde{\boldsymbol{\beta}})$. Using this constant function as an upper bound in Theorem 5, we can derive the corollary directly. □

Hence, our framework can be seen as a generalization of Gap Safe. Owing to this generalization, we can use a stronger problem-adaptive upper bound rather than a weak duality. In the next section, we derive specific regions for the *Lasso-Like* problem. Some regions for other problems are presented in the appendix.

# 4 Safe Region for Lasso-like Problem

In this section, we introduce a strong upper bound for the dual problems of Lasso and similar problems. The dome region derived from it can be seen as a generalization of Sasvi [14] and is narrower than Gap Safe region.

## 4.1 Norm-regularized Least Squares Problem and its Generalization

Norm-regularized least squares is an optimization problem and is formulated as $\underset{\boldsymbol{\beta} \in \mathbb{R}^d}{\text{minimize}} \ \frac{1}{2}\|\boldsymbol{y} - \boldsymbol{X}\boldsymbol{\beta}\|_2^2 + g(\boldsymbol{\beta})$ where $g$ is a norm. This is a subset of problem 2. Although this formulation includes Lasso [23], (overlapping) group-Lasso [26, 12], and ordered weighted L1 regression [8], the non-negative Lasso is not included. To unify them, we define the *Lasso-like* problem as follows:

$$\underset{\boldsymbol{\beta} \in \mathbb{R}^d}{\text{minimize}} \ \frac{1}{2}\|\boldsymbol{y} - \boldsymbol{X}\boldsymbol{\beta}\|_2^2 + g(\boldsymbol{\beta}), \tag{4}$$

where the problem satisfies all assumptions for Eq. (2) and $g$ satisfies

$$\forall a \geq 0, \boldsymbol{\beta} \in \mathbb{R}^d \ g(a\boldsymbol{\beta}) = ag(\boldsymbol{\beta}). \tag{5}$$

For the Lasso-like problem, the Fenchel conjugate functions of $f$ and $g$ are given as

$$f^\star(-\boldsymbol{\theta}) = \frac{1}{2}\|\boldsymbol{\theta}\|_2^2 - \boldsymbol{y}^\top\boldsymbol{\theta} \qquad g^\star(\boldsymbol{X}^\top\boldsymbol{\theta}) = \begin{cases} 0 & (\forall\boldsymbol{\beta} \ \boldsymbol{\theta}^\top\boldsymbol{X}\boldsymbol{\beta} - g(\boldsymbol{\beta}) \leq 0) \\ \infty & (\exists\boldsymbol{\beta} \ \boldsymbol{\theta}^\top\boldsymbol{X}\boldsymbol{\beta} - g(\boldsymbol{\beta}) > 0). \end{cases} \tag{6}$$

Note that $\{\boldsymbol{\theta} \mid g^\star(\boldsymbol{X}^\top\boldsymbol{\theta}) = 0\}$ is a closed convex set. Hence, the Lasso-like problem is a class of problems whose Fenchel–Rockafellar dual can be seen as a convex projection.

## 4.2 Proposed Dome Region for Lasso-like Problem

Using Theorem 5, we can construct a safe region by proposing an upper bound, $u(\boldsymbol{\theta})$. In this section, we propose a tight upper bound for Lasso-like problems. The direct expression of $f^\star$ in Eq. (6) is sufficiently simple. We only need an upper bound of $-g^\star$ to construct a simple region. The upper bound is given as follows.

**Lemma 7** *For Lasso-like problems* (4)*, for* $\forall\tilde{\boldsymbol{\beta}} \in \mathbb{R}^d$ *and* $\forall\boldsymbol{\theta} \in \mathbb{R}^n$*, we have*

$$D(\boldsymbol{\theta}) \leq -f^\star(-\boldsymbol{\theta}) + \inf_{a\geq 0} g(a\tilde{\boldsymbol{\beta}}) - \boldsymbol{\theta}^\top\boldsymbol{X}(a\tilde{\boldsymbol{\beta}}) = \begin{cases} -f^\star(-\boldsymbol{\theta}) & (g(\tilde{\boldsymbol{\beta}}) - \boldsymbol{\theta}^\top\boldsymbol{X}\tilde{\boldsymbol{\beta}} \geq 0) \\ -\infty & (g(\tilde{\boldsymbol{\beta}}) - \boldsymbol{\theta}^\top\boldsymbol{X}\tilde{\boldsymbol{\beta}} < 0). \end{cases} \tag{7}$$

The proof of Lemma 7 is given in the appendix. Using the right-hand side of Eq. (7) as an upper bound of $D$, we can construct a simple and safe region.

**Theorem 8** *Consider the Lasso-like problem described by Eq.* (4) *and its Fenchel–Rockafellar dual problem presented in Eq.* (3)*. Let* $\hat{\boldsymbol{\theta}}$ *be the dual optimal point. We assume that* $\tilde{\boldsymbol{\beta}} \in \mathbb{R}^d$ *and* $\tilde{\boldsymbol{\theta}} \in \text{dom}(D)$*. Then,* $\hat{\boldsymbol{\theta}}$ *is within the Dynamic Sasvi region, which is given as the intersection of a sphere and a half space:*

$$\mathcal{R}^{\text{DS}}(\tilde{\boldsymbol{\beta}}, \tilde{\boldsymbol{\theta}}) := \{\boldsymbol{\theta} \mid \frac{1}{2}\|\boldsymbol{\theta} - \tilde{\boldsymbol{\theta}}\|_2^2 + D(\tilde{\boldsymbol{\theta}}) \leq -f^\star(-\boldsymbol{\theta}) \wedge 0 \leq g(\tilde{\boldsymbol{\beta}}) - \boldsymbol{\theta}^\top\boldsymbol{X}\tilde{\boldsymbol{\beta}}\}$$

$$= \{\boldsymbol{\theta} \mid (\boldsymbol{y} - \boldsymbol{\theta})^\top(\tilde{\boldsymbol{\theta}} - \boldsymbol{\theta}) \leq 0 \wedge 0 \leq g(\tilde{\boldsymbol{\beta}}) - \boldsymbol{\theta}^\top\boldsymbol{X}\tilde{\boldsymbol{\beta}}\}.$$

The proof of Theorem 8 is given in the appendix. As proven in section 4.5, $\mathcal{R}^{\text{DS}}(\boldsymbol{\beta}^k, \boldsymbol{\theta}^k)$ converges to $\{\hat{\boldsymbol{\theta}}\}$ when $\lim_{k\to\infty} P(\boldsymbol{\beta}^k) - D(\boldsymbol{\theta}^k) = 0$. When we have good primal/dual feasible points, we can derive a very narrow safe region and eliminate almost all irrelevant features.

## 4.3 Relation to Sasvi

In this section, we show that safe screening with variational inequality (Sasvi) [14] is a special case of our screening rule. First, we review Sasvi. The target task of Sasvi is to minimize $\frac{1}{2}\|\lambda^{-1}\boldsymbol{y} - \boldsymbol{X}\boldsymbol{\beta}\|_2^2 + \|\boldsymbol{\beta}\|_1$ with many $\lambda$s. Sasvi is called "sequential" screening because it is designed to solve a sequence of problems with a sequence of penalty parameters, $\lambda_1 > \lambda_2 > \ldots > \lambda_M$. Let $\hat{\boldsymbol{\beta}}^{(\lambda)}$ and $\hat{\boldsymbol{\theta}}^{(\lambda)}$ be the optimal points of the primal problem and its Fenchel–Rockafellar dual problem, respectively. Because the dual problem can be interpreted as a projection from $\lambda^{-1}\boldsymbol{y}$ to a closed convex set, $\{\boldsymbol{\theta} \in \mathbb{R}^n \mid \|\boldsymbol{X}^\top\boldsymbol{\theta}\|_\infty \leq 1\}$, the following two variational inequalities hold:

$$0 \geq (\lambda_2^{-1}\boldsymbol{y} - \hat{\boldsymbol{\theta}}^{(\lambda_2)})^\top(\hat{\boldsymbol{\theta}}^{(\lambda_1)} - \hat{\boldsymbol{\theta}}^{(\lambda_2)}) \qquad 0 \geq (\lambda_1^{-1}\boldsymbol{y} - \hat{\boldsymbol{\theta}}^{(\lambda_1)})^\top(\hat{\boldsymbol{\theta}}^{(\lambda_2)} - \hat{\boldsymbol{\theta}}^{(\lambda_1)}). \tag{8}$$

Given $\hat{\boldsymbol{\theta}}^{(\lambda_1)}$, these inequalities provide a safe region for $\hat{\boldsymbol{\theta}}^{(\lambda_2)}$. Sasvi finds some zero elements of $\hat{\boldsymbol{\beta}}^{(\lambda_2)}$ based on this region. Note that $-\hat{\boldsymbol{\theta}}^{(\lambda_1)} = \boldsymbol{X}\hat{\boldsymbol{\beta}}^{(\lambda_1)} - \lambda_1^{-1}\boldsymbol{y}$ and $-\hat{\boldsymbol{\theta}}^{(\lambda_1)\top}\boldsymbol{X}\hat{\boldsymbol{\beta}}^{(\lambda_1)} = -g(\hat{\boldsymbol{\beta}}^{(\lambda_1)})$ are derived from Proposition 3. Then, we have

$$(\lambda_1^{-1}\boldsymbol{y} - \hat{\boldsymbol{\theta}}^{(\lambda_1)})^\top(\hat{\boldsymbol{\theta}}^{(\lambda_2)} - \hat{\boldsymbol{\theta}}^{(\lambda_1)}) = \hat{\boldsymbol{\theta}}^{(\lambda_2)\top}\boldsymbol{X}\hat{\boldsymbol{\beta}}^{(\lambda_1)} - g(\hat{\boldsymbol{\beta}}^{(\lambda_1)}).$$

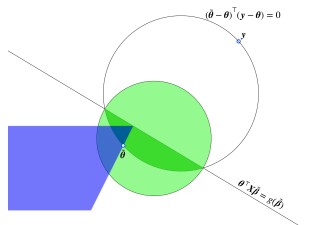 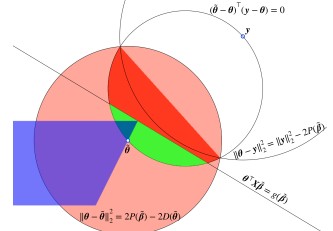 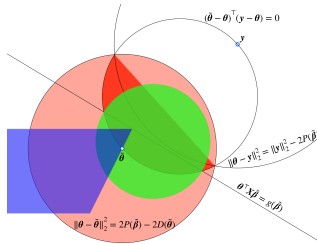

(a) Regions of dynamic Sasvi (dark green) and dynamic EDPP (light green).

(b) Regions of dynamic Sasvi (green), Gap Safe Sphere (light red) and Gap Safe Dome (dark red).

(c) Regions of dynamic EDPP (green), Gap Safe Sphere (light red), Gap Safe Dome (dark red).

Figure 1: Comparisons of various safe regions for Lasso. The blue region represents the feasible region.

Because $\lambda_2{}^{-1}\boldsymbol{y}$ in Eq. (8) corresponds to $\boldsymbol{y}$ in Theorem 8, $\mathcal{R}^{\mathrm{DS}}(\hat{\boldsymbol{\beta}}^{(\lambda_1)}, \hat{\boldsymbol{\theta}}^{(\lambda_1)})$ is equal to the Sasvi region. Hence, we can state that our method is a generalization of Sasvi. Our method requires primal/dual feasible points, unlike the original Sasvi, which requires the dual optimal point of a problem with another parameter. Such screening rules are called "dynamic" because they dynamically provide safe regions as iterative optimization proceeds. For this reason, we have labeled it as "Dynamic Sasvi." Owing to its "dynamic" property, Dynamic Sasvi eliminates almost all irrelevant features in the late step of the iterative optimization. As reported in [7], some sequential safe screening rules, including Sasvi, are not safe in practice because we cannot provide an exact solution for $\lambda_1$. Dynamic Sasvi overcomes this problem because it does not require the solution of a problem with another parameter.

## 4.4 Comparison to Gap Safe

Here, we show that the proposed method is stronger than Gap Safe Dome [7] and Gap Safe Sphere [7], [17] for Lasso-like problems. As shown in [7], for Lasso, the regions of the Gap Safe Dome and Gap Safe Sphere are the relaxation of the intersection of a sphere and the contra of another sphere:

$$\{\boldsymbol{\theta} \mid (\boldsymbol{y} - \boldsymbol{\theta})^\top (\tilde{\boldsymbol{\theta}} - \boldsymbol{\theta}) \leq 0 \wedge -f^\star(-\boldsymbol{\theta}) \leq P(\tilde{\boldsymbol{\beta}})\},$$

where $\tilde{\boldsymbol{\beta}}$ and $\tilde{\boldsymbol{\theta}}$ are primal/dual feasible vectors, respectively. $(\boldsymbol{y} - \hat{\boldsymbol{\theta}})^\top (\tilde{\boldsymbol{\theta}} - \hat{\boldsymbol{\theta}}) \leq 0$ is the variational inequality, and $-f^\star(-\hat{\boldsymbol{\theta}}) \leq P(\tilde{\boldsymbol{\beta}})$ is derived from weak duality. Using a simple transformation, we have

$$-f^\star(-\boldsymbol{\theta}) \leq P(\tilde{\boldsymbol{\beta}}) \iff -\frac{1}{2}\|\boldsymbol{\theta}\|_2^2 + \boldsymbol{y}^\top \boldsymbol{\theta} \leq \frac{1}{2}\|\boldsymbol{y} - \boldsymbol{X}\tilde{\boldsymbol{\beta}}\|_2^2 + g(\tilde{\boldsymbol{\beta}})$$

$$\iff -\frac{1}{2}\|\boldsymbol{y} - \boldsymbol{X}\tilde{\boldsymbol{\beta}} - \boldsymbol{\theta}\|_2^2 \leq g(\tilde{\boldsymbol{\beta}}) - \boldsymbol{\theta}^\top \boldsymbol{X}\tilde{\boldsymbol{\beta}}.$$

Hence, we have $0 \leq g(\tilde{\boldsymbol{\beta}}) - \boldsymbol{\theta}^\top \boldsymbol{X}\tilde{\boldsymbol{\beta}} \implies -f^\star(-\boldsymbol{\theta}) \leq P(\tilde{\boldsymbol{\beta}})$. This means that the region of dynamic Sasvi is a subset of the region of Gap Safe Dome. Our screening is always stronger than Gap Safe Dome and Gap Safe Sphere. Figure 1b shows the regions of Dynamic Sasvi, Gap Safe Dome, and Gap Safe Sphere.

## 4.5 Sphere Relaxation (Dynamic EDPP)

In some situations, even a dome region is too complicated to calculate $\bigcup_{\boldsymbol{\theta} \in \mathcal{R}} \partial g^\star(\boldsymbol{X}^\top \boldsymbol{\theta})$. For such cases, we propose using a minimum radius sphere that includes the dynamic Sasvi region. This method can be seen as a dynamic variant of the EDPP [24] because the EDPP is the minimum radius sphere relaxation of Sasvi.

**Theorem 9** *Consider the Lasso-like problem presented in Eq.* (4) *and its Fenchel–Rockafellar dual problem in Eq.* (3). *We assume that $\tilde{\boldsymbol{\beta}} \in \mathbb{R}^d$ and $\tilde{\boldsymbol{\theta}} \in \mathrm{dom}(D)$. If $n \geq 2$, the minimum radius sphere including $\mathcal{R}^{\mathrm{DS}}(\tilde{\boldsymbol{\beta}}, \tilde{\boldsymbol{\theta}})$ is*

$$\mathcal{R}^{\mathrm{DE}}(\tilde{\boldsymbol{\beta}}, \tilde{\boldsymbol{\theta}}) := \left\{\boldsymbol{\theta} \mid \left\|\boldsymbol{\theta} - \frac{1}{2}(\tilde{\boldsymbol{\theta}} + \boldsymbol{y}) + \alpha \boldsymbol{X}\tilde{\boldsymbol{\beta}}\right\|_2^2 \leq \frac{1}{4}\|\tilde{\boldsymbol{\theta}} - \boldsymbol{y}\|_2^2 - \alpha^2\|\boldsymbol{X}\tilde{\boldsymbol{\beta}}\|_2^2\right\}, \tag{9}$$

---

**Algorithm 1** Coordinate descent with Dynamic Sasvi for Lasso

---

1: **Input:** $\boldsymbol{X}, \boldsymbol{y}, \boldsymbol{\beta}^0, T, c, \epsilon$
2: Initialize $\tilde{\boldsymbol{\beta}} \leftarrow \boldsymbol{\beta}^0, \mathcal{A} \leftarrow [\![d]\!]$
3: **for** $t \in [\![T]\!]$ **do**
4:    **if** $t \mod c = 1$ **then**
5:       Compute $\tilde{\boldsymbol{\theta}} = \phi_{\mathcal{A}}(\tilde{\boldsymbol{\beta}})$
6:       **if** $P(\tilde{\boldsymbol{\beta}}) - D(\tilde{\boldsymbol{\theta}}) \le \frac{1}{2}\|\boldsymbol{y}\|_2^2 \epsilon$ **then**
7:          **break**
8:       **end if**
9:       $\mathcal{R} \leftarrow \mathcal{R}^{\mathrm{DS}}(\tilde{\boldsymbol{\beta}}, \tilde{\boldsymbol{\theta}})$
10:      $\mathcal{A} \leftarrow \{j \in \mathcal{A} : \max_{\boldsymbol{\theta} \in \mathcal{R}} |\boldsymbol{x}_j^\top \boldsymbol{\theta}| \ge 1\}$
11:      **for** $j \in [\![d]\!] - \mathcal{A}$ **do**
12:         $\tilde{\boldsymbol{\beta}}_j \leftarrow 0$
13:      **end for**
14:    **end if**
15:    **for** $j \in \mathcal{A}$ **do**
16:      $u \leftarrow \tilde{\boldsymbol{\beta}}_j \|\boldsymbol{x}_j\|_2^2 - \boldsymbol{x}_j^\top (\boldsymbol{X}\tilde{\boldsymbol{\beta}} - \boldsymbol{y})$
17:      $\tilde{\boldsymbol{\beta}}_j \leftarrow \frac{1}{\|\boldsymbol{x}_j\|_2^2} \mathrm{sign}(u) \max(0, |u| - 1)$
18:    **end for**
19: **end for**
20: **Output:** $\tilde{\boldsymbol{\beta}}$

---

*where*

$$\alpha = \max\left(0, \|\boldsymbol{X}\tilde{\boldsymbol{\beta}}\|_2^{-2}\left(\frac{1}{2}(\tilde{\boldsymbol{\theta}} + \boldsymbol{y})^\top \boldsymbol{X}\tilde{\boldsymbol{\beta}} - g(\tilde{\boldsymbol{\beta}})\right)\right).$$

The proof of Theorem 9 is provided in the appendix. Figures 1a and 1c show the dynamic EDPP region and other regions. To compare the Dynamic EDPP region and the Gap Safe Sphere region in general, we present the next theorem.

**Theorem 10** *Let $r$ be the radius of $\mathcal{R}^{\mathrm{DE}}(\tilde{\boldsymbol{\beta}}, \tilde{\boldsymbol{\theta}})$. We then have*

$$r < \sqrt{P(\tilde{\boldsymbol{\beta}}) - D(\tilde{\boldsymbol{\theta}})}.$$

The proof is given in the appendix. Because the radius of the Gap Safe Sphere region is $\sqrt{2(P(\tilde{\boldsymbol{\beta}}) - D(\tilde{\boldsymbol{\theta}}))}$, the squared radius of the Dynamic EDPP region is always less than half that of Gap Safe Sphere. Furthermore, Theorem 10 shows that $\mathcal{R}^{\mathrm{DE}}(\boldsymbol{\beta}^k, \boldsymbol{\theta}^k)$ and $\mathcal{R}^{\mathrm{DS}}(\boldsymbol{\beta}^k, \boldsymbol{\theta}^k)$ converge to $\{\hat{\boldsymbol{\theta}}\}$ when $\lim_{k\to\infty} P(\boldsymbol{\beta}^k) - D(\boldsymbol{\theta}^k) = 0$.

## 5 Implementation for Lasso

In this section, we provide a specific solver based on Theorem 8. Because the algorithm used to calculate $\bigcup_{\boldsymbol{\theta} \in \mathcal{R}} \partial g^\star(\boldsymbol{X}^\top \boldsymbol{\theta})$ depends on $g$, we introduce a Lasso solver as an example. We must choose an iterative solver to combine with the screening methods because they cannot estimate the solution alone. Although our methods can work with any iterative method, we use coordinate descent, which is recommended in [9].

### 5.1 Choice of a Dual Feasible Point

As shown in the previous section, $\lim_{t\to\infty} \mathcal{R}^{\mathrm{DS}}(\boldsymbol{\beta}^t, \boldsymbol{\theta}^t)$ converges to $\{\hat{\boldsymbol{\theta}}\}$ when $\lim_{t\to\infty} P(\boldsymbol{\beta}^t) - D(\boldsymbol{\theta}^t) = 0$ holds. Because the iterative solver provides a sequence of primal points whose $P(\boldsymbol{\beta}^t)$ converges to $D(\hat{\boldsymbol{\theta}})$, we only need a converging sequence of dual points to obtain a converging safe region. The next theorem provides such a sequence.

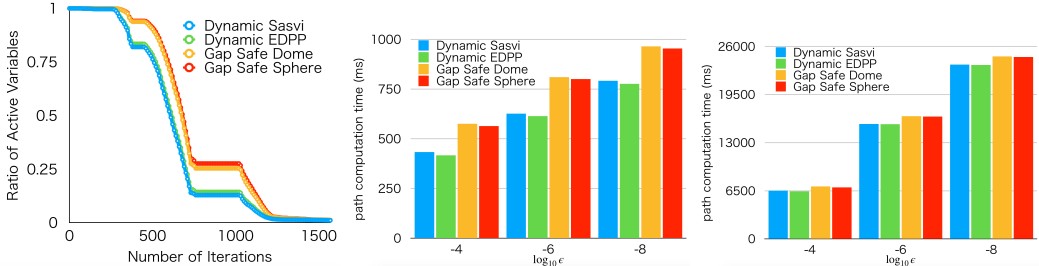

(a) Feature remaining rate (b) Computational time (Leukemia). (c) Computational time (RCV1).
(Leukemia).

Figure 2: (a): Feature remaining rate of each iteration for Lasso on leukemia (dense data with $n = 72, d = 7128$). (b) Average computational time of the Lasso path on subsampled leukemia (dense data with $n = 50, d = 7128$). (c): Average computational time of the Lasso path on subsampled RCV1 (sparse data with $n = 20000, d = 47236$).

Table 1: Logarithm of acceleration ratio. The larger values indicate a greater speed-up.

| Dataset | -log $\epsilon$ | Dynamic Sasvi | Dynamic EDPP | Gap Safe Dome | Gap Safe Sphere |
|---------|------|---------------|--------------|---------------|-----------------|
| Leukemia | 4 | $0.468 \pm 0.066$ | $0.487 \pm 0.066$ | $0.349 \pm 0.066$ | $0.358 \pm 0.060$ |
| | 6 | $0.828 \pm 0.072$ | $0.838 \pm 0.067$ | $0.719 \pm 0.074$ | $0.725 \pm 0.072$ |
| | 8 | $0.987 \pm 0.057$ | $0.997 \pm 0.056$ | $0.902 \pm 0.066$ | $0.907 \pm 0.066$ |
| RCV1 | 4 | $0.257 \pm 0.0056$ | $0.263 \pm 0.0078$ | $0.221 \pm 0.0081$ | $0.228 \pm 0.0089$ |
| | 6 | $0.373 \pm 0.0065$ | $0.374 \pm 0.0085$ | $0.345 \pm 0.0099$ | $0.346 \pm 0.0114$ |
| | 8 | $0.417 \pm 0.0065$ | $0.418 \pm 0.0079$ | $0.397 \pm 0.0086$ | $0.398 \pm 0.0092$ |

**Theorem 11** *(Converging $\boldsymbol{\theta}^t$) Consider the optimization problem in Eq. (4) with $g(\boldsymbol{\beta}) = \|\boldsymbol{\beta}\|_1$. Let $\hat{\boldsymbol{\beta}} \in \mathbb{R}^d$ and $\hat{\boldsymbol{\theta}} \in \mathbb{R}^n$ be the primal/dual solution. Assume that $\lim_{t \to \infty} \boldsymbol{\beta}^t = \hat{\boldsymbol{\beta}}$. Let us define $\phi : \mathbb{R}^d \to \mathbb{R}^n$ as*

$$\phi(\boldsymbol{\beta}) := \frac{1}{\max(1, \|\boldsymbol{X}^\top(\boldsymbol{y} - \boldsymbol{X}\boldsymbol{\beta})\|_\infty)}(\boldsymbol{y} - \boldsymbol{X}\boldsymbol{\beta}).$$

*Then, $\forall \boldsymbol{\beta}\ \phi(\boldsymbol{\beta}) \in \mathrm{dom}(D)$ and $\lim_{t \to \infty} \phi(\boldsymbol{\beta}^t) = \hat{\boldsymbol{\theta}}$ hold.*

**(Proof)** $\phi(\boldsymbol{\beta}) \in \mathrm{dom}(D)$ is directly derived from $\|\boldsymbol{X}^\top \phi(\boldsymbol{\beta})\|_\infty = \min(\|\boldsymbol{X}^\top(\boldsymbol{y} - \boldsymbol{X}\boldsymbol{\beta})\|_\infty, 1) \leq 1$. Because $\{\boldsymbol{y} - \boldsymbol{X}\boldsymbol{\beta}^t\}$ converges to $\boldsymbol{y} - \boldsymbol{X}\hat{\boldsymbol{\beta}} = \hat{\boldsymbol{\theta}}$, $\lim_{t \to \infty} \phi(\boldsymbol{\beta}^t) = \hat{\boldsymbol{\theta}}$ also holds. $\qquad\square$

If $\mathcal{A}$ is the set of features that is not yet eliminated, we can use

$$\phi_{\mathcal{A}}(\boldsymbol{\beta}) := \frac{1}{\max(1, \max_{j \in \mathcal{A}} \boldsymbol{x}_j^\top(\boldsymbol{y} - \boldsymbol{X}\boldsymbol{\beta}))}(\boldsymbol{X}\boldsymbol{\beta} - \boldsymbol{y})$$

instead of $\phi(\boldsymbol{\beta})$. Although $\phi_{\mathcal{A}}(\boldsymbol{\beta}) \in \mathrm{dom}(D)$ is not guaranteed, $\phi_{\mathcal{A}}(\boldsymbol{\beta})$ is guaranteed to satisfy all the constraints that are active in the dual solution. In other words, $\phi_{\mathcal{A}}(\boldsymbol{\beta})$ is in the domain of the dual problem of the small primal problem without eliminated features. Now, we can optimize the problem using the proposed screening method. The pseudocode is described in Algorithm 1. The direct expression of $\max_{\boldsymbol{\theta} \in \mathcal{R}^{\mathrm{DS}}(\tilde{\boldsymbol{\beta}}, \tilde{\boldsymbol{\theta}})} |\boldsymbol{x}_j^\top \boldsymbol{\theta}|$ is given in the appendix.

## 5.2 Computational Cost

In the screening part of Algorithm 1, only multiplying $\boldsymbol{X}$ or $\boldsymbol{X}^\top$ costs $O(nd)$. Because $\tilde{\boldsymbol{\theta}} = \phi_{\mathcal{A}}(\tilde{\boldsymbol{\beta}})$ is a linear combination of $\boldsymbol{X}\tilde{\boldsymbol{\beta}}$ and $\boldsymbol{y}$, we can obtain $\boldsymbol{X}^\top \boldsymbol{\theta}$ from $\boldsymbol{X}^\top \boldsymbol{X}\tilde{\boldsymbol{\beta}}$ and $\boldsymbol{X}^\top \boldsymbol{y}$. Because $\boldsymbol{X}^\top \boldsymbol{y}$ is constant, only the calculations of $\boldsymbol{X}\tilde{\boldsymbol{\beta}}$ and $\boldsymbol{X}^\top \boldsymbol{X}\tilde{\boldsymbol{\beta}}$ cost $O(nd)$. Hence, the screening cost is almost the same for all methods, which require $\boldsymbol{X}^\top \boldsymbol{X}\tilde{\boldsymbol{\beta}}$, including Gap Safe.

# 6 Experiments

In this section, we show the efficacy of the proposed methods using real-world data. We compared the proposed methods with Gap Safe Sphere and Gap Safe Dome [7, 17], which are state-of-the-art dynamic safe screening methods. All methods were run on a Macbook Air with a 1.1 GHz quad-core Intel Core i5 CPU with 16 GB of RAM. All methods were implemented in C++ using the Accelerate framework [1]. In this section, methods are evaluated on Lasso. Experiments on group Lasso are provided in the appendix.

## 6.1 Number of Screened Variables

First, we compared the number of screened variables among the four dynamic safe screening methods. We solved the Lasso problem using the leukemia dataset [2] [11] (dense data with 72 samples and 7128 features) and $\lambda = \frac{1}{100}\|\boldsymbol{X}^\top\boldsymbol{y}\|_\infty$. We used cyclic coordinate descent as the iterative algorithm and screened the variables for 10 iterations each. Figure 2a shows the ratio of the uneliminated features at each iteration. As guaranteed theoretically, we can see that Dynamic Sasvi eliminates more variables in earlier steps compared with both Gap Safe Dome and Gap Safe Sphere. The figure also shows that the Dynamic EDPP, which is a relaxed version of Dynamic Sasvi, eliminates almost the same number of features as Dynamic Sasvi.

## 6.2 Gains in the Computation of Lasso Paths

Next, we compared the computation time of the path of the Lasso solutions for various values of $\lambda$. We used $\lambda_j = 100^{-\frac{j}{99}}\|\boldsymbol{X}^\top\boldsymbol{y}\|_\infty$ ($j = 0, \ldots, 99$). We used the estimated primal solution for $\lambda_j$ with scaling as the initial vector in the solver for the problem with $\lambda_{j+1}$. The iterative solver stops when the duality gap is smaller than $\epsilon(P(\mathbf{0}) - D(\mathbf{0}))$. Note that $P(\mathbf{0}) - D(\mathbf{0})$ makes the stopping criterion independent of the data scale. We used the leukemia and RCV1 [3] [13] datasets (sparse data with 23149 samples and 47236 features). We subsampled the data 50 times and ran all the methods for the same 50 subsamples. The subsampled data size was 50 for leukemia and 20000 for RCV1. Figures 2b and 2c show the average computation times of the Lasso path for the leukemia and RCV1 datasets, respectively. For all settings, dynamic Sasvi and dynamic EDPP outperform Gap Safe Dome and Gap Safe Sphere. Table 1 lists the average values and standard deviations of the negative of the logarithm of the computational time ratio with respect to that for the same subsample without screening. The proposed methods are significantly faster than the Gap Safe methods.

# 7 Conclusion

In this study, we proposed a framework for safe screening based on the Fenchel–Rockafellar duality and derived Dynamic Sasvi and Dynamic EDPP, which are specific safe screening methods for Lasso-like problems. Dynamic Sasvi and Dynamic EDPP can be regarded as dynamic feature elimination variants of Sasvi and EDPP, respectively. We proved that Dynamic Sasvi always eliminates more features than both Gap Safe Sphere and Gap Safe Dome. Dynamic EDPP is based on the sphere relaxation of the Dynamic Sasvi region and eliminates almost the same number of features as Dynamic Sasvi. We also showed experimentally that the computational costs of the proposed methods are lower than those of Gap Safe Sphere and Gap Safe Dome.

## Acknowledgement

MY was supported by MEXT KAKENHI 20H04243 and partly supported by MEXT KAKENHI 21H04874.

---

[1] Source codes are in `https://github.com/k8127i/2021DynamicSasvi`

[2] `https://leo.ugr.es/elvira/DBCRepository/Leukemia/ALLAML.html`

[3] `https://scikit-learn.org/0.18/datasets/rcv1.html`

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
