# A Proof of Theorems

## A.1 Proof of Theorem 2

**(Proof)** According to Theorem 15.23 [2], since we have assumed that

$$\exists \boldsymbol{\beta} \in \mathrm{relint}(\mathrm{dom}(g)) \text{ s.t. } \boldsymbol{X}\boldsymbol{\beta} \in \mathrm{relint}(\mathrm{dom}(f)),$$

i.e., $\mathrm{relint}(\mathrm{dom}(f)) \cap \boldsymbol{X}\mathrm{relint}(\mathrm{dom}(g))$ is not empty, we have

$$\inf_{\boldsymbol{\beta}\in\mathbb{R}^d} f(\boldsymbol{X}\boldsymbol{\beta}) + g(\boldsymbol{\beta}) = \max_{\boldsymbol{\theta}\in\mathbb{R}^n} -f^\star(-\boldsymbol{\theta}) - g^\star(\boldsymbol{X}^\top\boldsymbol{\theta}).$$

In addition, we have assumed the existence of the optimal point. Hence, we have

$$\min_{\boldsymbol{\beta}\in\mathbb{R}^d} f(\boldsymbol{X}\boldsymbol{\beta}) + g(\boldsymbol{\beta}) = \max_{\boldsymbol{\theta}\in\mathbb{R}^n} -f^\star(-\boldsymbol{\theta}) - g^\star(\boldsymbol{X}^\top\boldsymbol{\theta}).$$

$\square$

## A.2 Proof of Lemma 7

**(Proof)** Based on Fenchel–Young inequality in Eq. (1), we have $-g^\star(\boldsymbol{X}^\top\boldsymbol{\theta}) \le g(k\tilde{\boldsymbol{\beta}}) - \boldsymbol{\theta}^\top\boldsymbol{X}(k\tilde{\boldsymbol{\beta}})$. Hence, we have

$$D(\boldsymbol{\theta}) \le -f^\star(-\boldsymbol{\theta}) + \inf_{k\ge 0} g(k\tilde{\boldsymbol{\beta}}) - \boldsymbol{\theta}^\top\boldsymbol{X}(k\tilde{\boldsymbol{\beta}}).$$

Based on Eq. (5), we have

$$
\begin{aligned}
D(\boldsymbol{\theta}) &\le -f^\star(-\boldsymbol{\theta}) + \inf_{k\ge 0} g(k\tilde{\boldsymbol{\beta}}) - \boldsymbol{\theta}^\top\boldsymbol{X}(k\tilde{\boldsymbol{\beta}}) \\
&= -f^\star(-\boldsymbol{\theta}) + \inf_{k\ge 0} k(g(\tilde{\boldsymbol{\beta}}) - \boldsymbol{\theta}^\top\boldsymbol{X}\tilde{\boldsymbol{\beta}}) \\
&= \begin{cases} -f^\star(-\boldsymbol{\theta}) & (g(\tilde{\boldsymbol{\beta}}) - \boldsymbol{\theta}^\top\boldsymbol{X}\tilde{\boldsymbol{\beta}} \ge 0) \\ -\infty & (g(\tilde{\boldsymbol{\beta}}) - \boldsymbol{\theta}^\top\boldsymbol{X}\tilde{\boldsymbol{\beta}} < 0). \end{cases}
\end{aligned}
$$

$\square$

## A.3 Proof of Theorem 8

**(Proof)** According to Theorem 5 and Lemma 7, we can easily obtain

$$\hat{\boldsymbol{\theta}} \in \mathcal{R}^{\mathrm{DS}}(\tilde{\boldsymbol{\beta}},\tilde{\boldsymbol{\theta}}) := \{\boldsymbol{\theta} \mid \frac{1}{2}\|\boldsymbol{\theta}-\tilde{\boldsymbol{\theta}}\|_2^2 + D(\tilde{\boldsymbol{\theta}}) \le \begin{cases} -f^\star(-\boldsymbol{\theta}) & (g(\tilde{\boldsymbol{\beta}}) - \boldsymbol{\theta}^\top\boldsymbol{X}\tilde{\boldsymbol{\beta}} \ge 0) \\ -\infty & (g(\tilde{\boldsymbol{\beta}}) - \boldsymbol{\theta}^\top\boldsymbol{X}\tilde{\boldsymbol{\beta}} < 0). \end{cases}\}.$$

Since $\tilde{\boldsymbol{\theta}}$ is in $\mathrm{dom}(D)$, $g^\star(\boldsymbol{X}^\top\tilde{\boldsymbol{\theta}})$ always be 0 and $D(\tilde{\boldsymbol{\theta}})$ never be $-\infty$. Hence we have

$$
\begin{aligned}
&\frac{1}{2}\|\boldsymbol{\theta}-\tilde{\boldsymbol{\theta}}\|_2^2 + D(\tilde{\boldsymbol{\theta}}) \le \begin{cases} -f^\star(-\boldsymbol{\theta}) & (g(\tilde{\boldsymbol{\beta}}) - \boldsymbol{\theta}^\top\boldsymbol{X}\tilde{\boldsymbol{\beta}} \ge 0) \\ -\infty & (g(\tilde{\boldsymbol{\beta}}) - \boldsymbol{\theta}^\top\boldsymbol{X}\tilde{\boldsymbol{\beta}} < 0). \end{cases} \\
\iff &\frac{1}{2}\|\boldsymbol{\theta}-\tilde{\boldsymbol{\theta}}\|_2^2 + D(\tilde{\boldsymbol{\theta}}) \le -f^\star(-\boldsymbol{\theta}) \wedge 0 \le g(\tilde{\boldsymbol{\beta}}) - \boldsymbol{\theta}^\top\boldsymbol{X}\tilde{\boldsymbol{\beta}} \\
\iff &\frac{1}{2}\|\boldsymbol{\theta}-\tilde{\boldsymbol{\theta}}\|_2^2 - \frac{1}{2}\|\tilde{\boldsymbol{\theta}}\|_2^2 + \boldsymbol{y}^\top\tilde{\boldsymbol{\theta}} \le -\frac{1}{2}\|\boldsymbol{\theta}\|_2^2 + \boldsymbol{y}^\top\boldsymbol{\theta} \wedge 0 \le g(\tilde{\boldsymbol{\beta}}) - \boldsymbol{\theta}^\top\boldsymbol{X}\tilde{\boldsymbol{\beta}} \\
\iff &(\boldsymbol{y}-\boldsymbol{\theta})^\top(\tilde{\boldsymbol{\theta}}-\boldsymbol{\theta}) \le 0 \wedge 0 \le g(\tilde{\boldsymbol{\beta}}) - \boldsymbol{\theta}^\top\boldsymbol{X}\tilde{\boldsymbol{\beta}}.
\end{aligned}
$$

$\square$

## A.4 Proof of Theorem 9

**Proof of $\mathcal{R}^{\mathrm{DS}}(\tilde{\boldsymbol{\beta}},\tilde{\boldsymbol{\theta}}) \subset \mathcal{R}^{\mathrm{DE}}(\tilde{\boldsymbol{\beta}},\tilde{\boldsymbol{\theta}})$:**

**(Proof)** Since $\alpha \geq 0$, we have

$$
\begin{aligned}
\mathcal{R}^{\mathrm{DS}}(\tilde{\boldsymbol{\beta}}, \tilde{\boldsymbol{\theta}}) =& \{\boldsymbol{\theta} \mid \|\boldsymbol{\theta} - \frac{1}{2}(\tilde{\boldsymbol{\theta}} + \boldsymbol{y})\|_2^2 \leq \frac{1}{4}\|(\tilde{\boldsymbol{\theta}} - \boldsymbol{y})\|_2^2 \wedge 0 \leq g(\tilde{\boldsymbol{\beta}}) - \boldsymbol{\theta}^\top \boldsymbol{X}\tilde{\boldsymbol{\beta}}\} \\
\subset& \{\boldsymbol{\theta} \mid \|\boldsymbol{\theta} - \frac{1}{2}(\tilde{\boldsymbol{\theta}} + \boldsymbol{y})\|_2^2 \leq \frac{1}{4}\|(\tilde{\boldsymbol{\theta}} - \boldsymbol{y})\|_2^2 + 2\alpha(g(\tilde{\boldsymbol{\beta}}) - \boldsymbol{\theta}^\top \boldsymbol{X}\tilde{\boldsymbol{\beta}})\} \\
=& \{\boldsymbol{\theta} \mid \|\boldsymbol{\theta}\|_2^2 - \boldsymbol{\theta}^\top(\tilde{\boldsymbol{\theta}} + \boldsymbol{y}) + 2\alpha\boldsymbol{\theta}^\top \boldsymbol{X}\tilde{\boldsymbol{\beta}} \leq \frac{1}{4}\|(\tilde{\boldsymbol{\theta}} - \boldsymbol{y})\|_2^2 - \frac{1}{4}\|\tilde{\boldsymbol{\theta}} + \boldsymbol{y}\|_2^2 + 2\alpha g(\tilde{\boldsymbol{\beta}})\} \\
=& \{\boldsymbol{\theta} \mid \|\boldsymbol{\theta} - \frac{1}{2}(\tilde{\boldsymbol{\theta}} + \boldsymbol{y}) + \alpha\boldsymbol{X}\tilde{\boldsymbol{\beta}}\|_2^2 \leq \frac{1}{4}\|(\tilde{\boldsymbol{\theta}} - \boldsymbol{y})\|_2^2 - \frac{1}{4}\|\tilde{\boldsymbol{\theta}} + \boldsymbol{y}\|_2^2 + \|\frac{1}{2}(\tilde{\boldsymbol{\theta}} + \boldsymbol{y}) - \alpha\boldsymbol{X}\tilde{\boldsymbol{\beta}}\|_2^2 + 2\alpha g(\tilde{\boldsymbol{\beta}})\} \\
=& \{\boldsymbol{\theta} \mid \|\boldsymbol{\theta} - \frac{1}{2}(\tilde{\boldsymbol{\theta}} + \boldsymbol{y}) + \alpha\boldsymbol{X}\tilde{\boldsymbol{\beta}}\|_2^2 \leq \frac{1}{4}\|(\tilde{\boldsymbol{\theta}} - \boldsymbol{y})\|_2^2 + \alpha^2\|\boldsymbol{X}\tilde{\boldsymbol{\beta}}\|_2^2 - \alpha(\tilde{\boldsymbol{\theta}} + \boldsymbol{y})^\top \boldsymbol{X}\tilde{\boldsymbol{\beta}} + 2\alpha g(\tilde{\boldsymbol{\beta}})\}.
\end{aligned}
$$

And by $\alpha \in \{0, \frac{1}{\|\boldsymbol{X}\tilde{\boldsymbol{\beta}}\|_2^2}(\frac{1}{2}(\tilde{\boldsymbol{\theta}} + \boldsymbol{y})^\top \boldsymbol{X}\tilde{\boldsymbol{\beta}} - g(\tilde{\boldsymbol{\beta}}))\}$, we have

$$
\alpha^2\|\boldsymbol{X}\tilde{\boldsymbol{\beta}}\|_2^2 - \alpha(\tilde{\boldsymbol{\theta}} + \boldsymbol{y})^\top \boldsymbol{X}\tilde{\boldsymbol{\beta}} + 2\alpha g(\tilde{\boldsymbol{\beta}}) = -\alpha^2\|\boldsymbol{X}\tilde{\boldsymbol{\beta}}\|_2^2.
$$

Hence,

$$
\begin{aligned}
\mathcal{R}^{\mathrm{DS}}(\tilde{\boldsymbol{\beta}}, \tilde{\boldsymbol{\theta}}) \subset& \{\boldsymbol{\theta} \mid \|\boldsymbol{\theta} - \frac{1}{2}(\tilde{\boldsymbol{\theta}} + \boldsymbol{y}) + \alpha\boldsymbol{X}\tilde{\boldsymbol{\beta}}\|_2^2 \leq \frac{1}{4}\|(\tilde{\boldsymbol{\theta}} - \boldsymbol{y})\|_2^2 - \alpha^2\|\boldsymbol{X}\tilde{\boldsymbol{\beta}}\|_2^2\} \\
=& \mathcal{R}^{\mathrm{DE}}(\tilde{\boldsymbol{\beta}}, \tilde{\boldsymbol{\theta}})
\end{aligned}
$$

holds. $\qquad\square$

### Proof of minimality of the radius:

**(Proof)** Let $\boldsymbol{v} \in \mathbb{R}^n$ be a vector which satisfies $\boldsymbol{v}^\top \boldsymbol{X}\tilde{\boldsymbol{\beta}} = 0$ and $\boldsymbol{v}^\top \boldsymbol{v} = 1$. Note that such a vector exists if $n \geq 2$. Then, we have $\frac{1}{2}(\tilde{\boldsymbol{\theta}} + \boldsymbol{y}) - \alpha\boldsymbol{X}\tilde{\boldsymbol{\beta}} \pm r\boldsymbol{v} \in \mathcal{R}^{\mathrm{DS}}(\tilde{\boldsymbol{\beta}}, \tilde{\boldsymbol{\theta}})$ because

$$
(\frac{1}{2}(\tilde{\boldsymbol{\theta}} + \boldsymbol{y}) - \alpha\boldsymbol{X}\tilde{\boldsymbol{\beta}} \pm r\boldsymbol{v})^\top \boldsymbol{X}\tilde{\boldsymbol{\beta}} = \frac{1}{2}(\tilde{\boldsymbol{\theta}} + \boldsymbol{y})^\top \boldsymbol{X}\tilde{\boldsymbol{\beta}} - \max(0, \frac{1}{2}(\tilde{\boldsymbol{\theta}} + \boldsymbol{y})^\top \boldsymbol{X}\tilde{\boldsymbol{\beta}} - g(\tilde{\boldsymbol{\beta}})) \leq g(\tilde{\boldsymbol{\beta}})
$$

and

$$
\|\frac{1}{2}(\tilde{\boldsymbol{\theta}} + \boldsymbol{y}) - \alpha\boldsymbol{X}\tilde{\boldsymbol{\beta}} \pm r\boldsymbol{v} - \frac{1}{2}(\tilde{\boldsymbol{\theta}} + \boldsymbol{y})\|_2^2 = \| - \alpha\boldsymbol{X}\tilde{\boldsymbol{\beta}} \pm r\boldsymbol{v}\|_2^2 = \frac{1}{4}\|(\tilde{\boldsymbol{\theta}} - \boldsymbol{y})\|_2^2
$$

hold. Since the distance between these two points is $2r$, the radius of a sphere which includes $\mathcal{R}^{\mathrm{DS}}(\tilde{\boldsymbol{\beta}}, \tilde{\boldsymbol{\theta}})$ can not be smaller than $r$. $\qquad\square$

## A.5 Proof of Theorem 10

**(Proof)** Let us denote $\frac{1}{\|\boldsymbol{X}\tilde{\boldsymbol{\beta}}\|_2}(\boldsymbol{y}^\top \boldsymbol{X}\tilde{\boldsymbol{\beta}} - g(\tilde{\boldsymbol{\beta}}))$ by $\gamma$. If $\frac{1}{2}\|\tilde{\boldsymbol{\theta}} - \boldsymbol{y}\|_2$ is smaller than $\gamma$, we have

$$
\begin{aligned}
r^2 - (P(\tilde{\boldsymbol{\beta}}) - D(\tilde{\boldsymbol{\theta}})) \leq& (\frac{1}{4}\|\tilde{\boldsymbol{\theta}} - \boldsymbol{y}\|_2^2 - (\gamma - \frac{1}{2}\|\tilde{\boldsymbol{\theta}} - \boldsymbol{y}\|_2)^2) - (P(\tilde{\boldsymbol{\beta}}) - D(\tilde{\boldsymbol{\theta}})) \\
=& -\gamma^2 + \gamma\|\tilde{\boldsymbol{\theta}} - \boldsymbol{y}\|_2 - \frac{1}{2}\|\boldsymbol{X}\tilde{\boldsymbol{\beta}}\|_2^2 + \|\boldsymbol{X}\tilde{\boldsymbol{\beta}}\|_2\gamma - \frac{1}{2}\|\tilde{\boldsymbol{\theta}} - \boldsymbol{y}\|_2^2 \\
=& -\frac{1}{2}(\|\boldsymbol{X}\tilde{\boldsymbol{\beta}}\|_2 - \gamma)^2 - \frac{1}{2}(\|\tilde{\boldsymbol{\theta}} - \boldsymbol{y}\|_2 - \gamma)^2 \leq 0
\end{aligned}
$$

If $\gamma$ is smaller than $\frac{1}{2}\|\tilde{\boldsymbol{\theta}} - \boldsymbol{y}\|_2$, we have

$$
\begin{aligned}
r^2 - (P(\tilde{\boldsymbol{\beta}}) - D(\tilde{\boldsymbol{\theta}})) \leq& \frac{1}{4}\|\tilde{\boldsymbol{\theta}} - \boldsymbol{y}\|_2^2 - (\frac{1}{2}\|\boldsymbol{X}\tilde{\boldsymbol{\beta}}\|_2^2 - \|\boldsymbol{X}\tilde{\boldsymbol{\beta}}\|_2\gamma + \frac{1}{2}\|\tilde{\boldsymbol{\theta}} - \boldsymbol{y}\|_2^2) \\
\leq& -\frac{1}{4}\|\boldsymbol{X}\tilde{\boldsymbol{\beta}}\|_2^2 - \frac{1}{4}(\|\boldsymbol{X}\tilde{\boldsymbol{\beta}}\|_2 - \|\tilde{\boldsymbol{\theta}} - \boldsymbol{y}\|_2)^2 \leq 0
\end{aligned}
$$

Hence, we have

$$
r < \sqrt{P(\tilde{\boldsymbol{\beta}}) - D(\tilde{\boldsymbol{\theta}})}
$$

$\qquad\square$

# B   Direct Expression of $\max_{\boldsymbol{\theta}\in\mathcal{R}^{\mathrm{DS}}(\tilde{\boldsymbol{\beta}},\tilde{\boldsymbol{\theta}})} \boldsymbol{x}_j^\top \boldsymbol{\theta}$

Let $r = \frac{1}{2}\|\tilde{\boldsymbol{\theta}} - \boldsymbol{y}\|_2$ and $\boldsymbol{\theta}_o = \frac{1}{2}(\tilde{\boldsymbol{\theta}} + \boldsymbol{y})$.

If $(\boldsymbol{\theta}_o + \frac{r}{\|\boldsymbol{x}_j\|_2}\boldsymbol{x}_j)^\top \boldsymbol{X}\tilde{\boldsymbol{\beta}} \leq g(\tilde{\boldsymbol{\beta}})$, $\operatorname{argmax}_{\boldsymbol{\theta}\in\mathcal{R}^{\mathrm{DS}}(\tilde{\boldsymbol{\beta}},\tilde{\boldsymbol{\theta}})} \boldsymbol{x}_j^\top \boldsymbol{\theta} = \boldsymbol{\theta}_o + \frac{r}{\|\boldsymbol{x}_j\|_2}\boldsymbol{x}_j$ and $\max_{\boldsymbol{\theta}\in\mathcal{R}^{\mathrm{DS}}(\tilde{\boldsymbol{\beta}},\tilde{\boldsymbol{\theta}})} \boldsymbol{x}_j^\top \boldsymbol{\theta} = \boldsymbol{x}_j^\top \boldsymbol{\theta}_o + r\|\boldsymbol{x}_j\|_2$.

If $(\boldsymbol{\theta}_o + \frac{r}{\|\boldsymbol{x}_j\|_2}\boldsymbol{x}_j)^\top \boldsymbol{X}\tilde{\boldsymbol{\beta}} > g(\tilde{\boldsymbol{\beta}})$, the constraint $\boldsymbol{\theta}^\top \boldsymbol{X}\tilde{\boldsymbol{\beta}} \leq g(\tilde{\boldsymbol{\beta}})$ guaranteed to be active at the solution. Hence,

$$
\max_{\boldsymbol{\theta}\in\mathcal{R}^{\mathrm{DS}}(\tilde{\boldsymbol{\beta}},\tilde{\boldsymbol{\theta}})} \boldsymbol{x}_j^\top \boldsymbol{\theta}
$$

$$
= \max_{\|\boldsymbol{\theta}-\boldsymbol{\theta}_o\|_2^2 \leq r^2 \wedge \boldsymbol{\theta}^\top \boldsymbol{X}\tilde{\boldsymbol{\beta}}=g(\tilde{\boldsymbol{\beta}})} \boldsymbol{x}_j^\top \boldsymbol{\theta}
$$

$$
= \boldsymbol{x}_j^\top \boldsymbol{\theta}_o + \frac{\boldsymbol{x}_j^\top \boldsymbol{X}\tilde{\boldsymbol{\beta}}}{\|\boldsymbol{X}\tilde{\boldsymbol{\beta}}\|_2^2}(g(\tilde{\boldsymbol{\beta}}) - \boldsymbol{\theta}_o^\top \boldsymbol{X}\tilde{\boldsymbol{\beta}}) + \max_{\|\boldsymbol{\theta}'\|_2^2 \leq r^2 \wedge \boldsymbol{\theta}'^\top \boldsymbol{X}\tilde{\boldsymbol{\beta}}=g(\tilde{\boldsymbol{\beta}})-\boldsymbol{\theta}_o^\top \boldsymbol{X}\tilde{\boldsymbol{\beta}}} \left(\boldsymbol{x}_j - \frac{\boldsymbol{x}_j^\top \boldsymbol{X}\tilde{\boldsymbol{\beta}}}{\|\boldsymbol{X}\tilde{\boldsymbol{\beta}}\|_2^2}\boldsymbol{X}\tilde{\boldsymbol{\beta}}\right)^\top \boldsymbol{\theta}'
$$

$$
= \boldsymbol{x}_j^\top \boldsymbol{\theta}_o + \frac{\boldsymbol{x}_j^\top \boldsymbol{X}\tilde{\boldsymbol{\beta}}}{\|\boldsymbol{X}\tilde{\boldsymbol{\beta}}\|_2^2}(g(\tilde{\boldsymbol{\beta}}) - \boldsymbol{\theta}_o^\top \boldsymbol{X}\tilde{\boldsymbol{\beta}}) + \left\|\boldsymbol{x}_j - \frac{\boldsymbol{x}_j^\top \boldsymbol{X}\tilde{\boldsymbol{\beta}}}{\|\boldsymbol{X}\tilde{\boldsymbol{\beta}}\|_2^2}\boldsymbol{X}\tilde{\boldsymbol{\beta}}\right\|_2 \sqrt{r^2 - \frac{1}{\|\boldsymbol{X}\tilde{\boldsymbol{\beta}}\|_2^2}(g(\tilde{\boldsymbol{\beta}}) - \boldsymbol{\theta}_o^\top \boldsymbol{X}\tilde{\boldsymbol{\beta}})^2}.
$$

Let $\delta = g(\tilde{\boldsymbol{\beta}}) - \boldsymbol{\theta}_o^\top \boldsymbol{X}\tilde{\boldsymbol{\beta}}$. Then,

$$
\max_{\boldsymbol{\theta}\in\mathcal{R}^{\mathrm{DS}}(\tilde{\boldsymbol{\beta}},\tilde{\boldsymbol{\theta}})} \boldsymbol{x}_j^\top \boldsymbol{\theta} = \begin{cases} \boldsymbol{x}_j^\top \boldsymbol{\theta}_o + r\|\boldsymbol{x}_j\|_2 & (\frac{r}{\|\boldsymbol{x}_j\|_2}\boldsymbol{x}_j^\top \boldsymbol{X}\tilde{\boldsymbol{\beta}} \leq \delta) \\ \boldsymbol{x}_j^\top \boldsymbol{\theta}_o + \frac{\boldsymbol{x}_j^\top \boldsymbol{X}\tilde{\boldsymbol{\beta}}}{\|\boldsymbol{X}\tilde{\boldsymbol{\beta}}\|_2^2}\delta + \left\|\boldsymbol{x}_j - \frac{\boldsymbol{x}_j^\top \boldsymbol{X}\tilde{\boldsymbol{\beta}}}{\|\boldsymbol{X}\tilde{\boldsymbol{\beta}}\|_2^2}\boldsymbol{X}\tilde{\boldsymbol{\beta}}\right\|_2 \sqrt{r^2 - \frac{1}{\|\boldsymbol{X}\tilde{\boldsymbol{\beta}}\|_2^2}\delta^2} & (\frac{r}{\|\boldsymbol{x}_j\|_2}\boldsymbol{x}_j^\top \boldsymbol{X}\tilde{\boldsymbol{\beta}} > \delta) \end{cases}
$$

# C   Regions for other problems

According to Theorem 5, we can construct simple safe region by constructing simple upper bound of $D(\boldsymbol{\theta})$. Herein, we introduce upper bounds for some non Lasso-like problems. All upper bounds in this section are stronger than weak duality.

**Elastic-Net**: Consider the following problem:

$$
\underset{\boldsymbol{\beta}\in\mathbb{R}^d}{\operatorname{minimize}} \ \frac{1}{2}\|\boldsymbol{y} - \boldsymbol{X}\boldsymbol{\beta}\|_2^2 + g(\boldsymbol{\beta}),
$$

where

$$
g(\boldsymbol{\beta}) = \|\boldsymbol{\beta}\|_1 + \frac{\alpha}{2}\|\boldsymbol{\beta}\|_2^2
$$

and $\alpha > 0$. Then, for $\forall\tilde{\boldsymbol{\beta}}$, we have

$$
D(\boldsymbol{\theta}) \leq -f^\star(-\boldsymbol{\theta}) + \inf_{k\geq 0} g(k\tilde{\boldsymbol{\beta}}) - \boldsymbol{\theta}^\top \boldsymbol{X}(k\tilde{\boldsymbol{\beta}})
$$

$$
= -f^\star(-\boldsymbol{\theta}) + \inf_{k\geq 0} k(\|\tilde{\boldsymbol{\beta}}\|_1 - \boldsymbol{\theta}^\top \boldsymbol{X}\tilde{\boldsymbol{\beta}}) + \frac{\alpha k^2}{2}\|\tilde{\boldsymbol{\beta}}\|_2^2
$$

$$
= \begin{cases} -f^\star(-\boldsymbol{\theta}) & (h(\boldsymbol{\theta};\tilde{\boldsymbol{\beta}}) \geq 0) \\ -f^\star(-\boldsymbol{\theta}) - \frac{h(\boldsymbol{\theta};\tilde{\boldsymbol{\beta}})^2}{2\alpha\|\tilde{\boldsymbol{\beta}}\|_2^2} & (h(\boldsymbol{\theta};\tilde{\boldsymbol{\beta}}) < 0), \end{cases}
$$

where

$$
h(\boldsymbol{\theta};\tilde{\boldsymbol{\beta}}) = \|\tilde{\boldsymbol{\beta}}\|_1 - \boldsymbol{\theta}^\top \boldsymbol{X}\tilde{\boldsymbol{\beta}}.
$$

**General $g$:** Except for Elastic-Net, there are many regularizers that do not satisfy Eq. (5), e.g., squared L1 regularization. In addition, the dual problem of SVM can be seen as the regularized least squares. In those cases, we propose using the upper bound

$$
D(\boldsymbol{\theta}) \leq -\frac{1}{2}\|\boldsymbol{\theta} - \boldsymbol{y}\|_2^2 + \frac{1}{2}\|\boldsymbol{y}\|_2^2 + g(\tilde{\boldsymbol{\beta}}) - \boldsymbol{\theta}^\top \boldsymbol{X}\tilde{\boldsymbol{\beta}}.
$$

This is based on the Fenchel–Young inequality for $g$ and Eq. (6).

**General $f$:** Here, we extend $f$ to a more general setup, e.g., the logistic loss. Assume that $g$ satisfies Eq. (5). In those cases, we propose using

$$D(\boldsymbol{\theta}) \leq f(\boldsymbol{X}\boldsymbol{\beta}) + \boldsymbol{\theta}^\top \boldsymbol{X}\boldsymbol{\beta} + \inf_{k \geq 0} g(k\tilde{\boldsymbol{\beta}}) - \boldsymbol{\theta}^\top \boldsymbol{X}(k\tilde{\boldsymbol{\beta}}),$$

$$= f(\boldsymbol{X}\boldsymbol{\beta}) + \boldsymbol{\theta}^\top \boldsymbol{X}\boldsymbol{\beta} + \begin{cases} 0 & (g(\tilde{\boldsymbol{\beta}}) - \boldsymbol{\theta}^\top \boldsymbol{X}\tilde{\boldsymbol{\beta}} \geq 0) \\ -\infty & (g(\tilde{\boldsymbol{\beta}}) - \boldsymbol{\theta}^\top \boldsymbol{X}\tilde{\boldsymbol{\beta}} < 0) \end{cases}.$$

This is based on the Fenchel–Young inequality and Eq. (5).

# D  Group Lasso

In this section, we show the empirical results of screening methods on non-overlapping group Lasso. We consider an optimization problem, formulated as

$$\underset{\boldsymbol{\beta} \in \mathbb{R}^d}{\text{minimize}} \ \frac{1}{2}\|\boldsymbol{y} - \boldsymbol{X}\boldsymbol{\beta}\|_2^2 + \sum_{s=1}^{M} \sqrt{\sum_{j=l_{s-1}+1}^{l_s} \beta_j^2},$$

where $0 = l_0 < l_1 < \ldots < l_M = d$. The Fenchel–Rockafellar dual problem is

$$\underset{\boldsymbol{\theta} \in \mathcal{G}}{\text{maximize}} \ -\frac{1}{2}\|\boldsymbol{\theta}\|_2^2 + \boldsymbol{y}^\top \boldsymbol{\theta},$$

where $\mathcal{G} = \{\boldsymbol{\theta} \in \mathbb{R}^n \mid \forall s \in \{1, 2, \ldots, M\} \ \sum_{j=l_{s-1}+1}^{l_s} (\boldsymbol{x}_j^\top \boldsymbol{\theta})^2 \leq 1\}$.

## D.1  Screening rule

Since $\partial g^\star(\boldsymbol{X}^\top \boldsymbol{\theta}) \subset \{\boldsymbol{\beta} \mid \boldsymbol{\beta}_{l_{s-1}+1} = \boldsymbol{\beta}_{l_{s-1}+2} = \ldots = \boldsymbol{\beta}_{l_s} = 0\} \iff \sum_{j=l_{s-1}+1}^{l_s} (\boldsymbol{x}_j^\top \boldsymbol{\theta})^2 < 1$, we have

$$\hat{\boldsymbol{\theta}} \in \mathcal{R} \wedge \max_{\boldsymbol{\theta} \in \mathcal{R}} \sum_{j=l_{s-1}+1}^{l_s} (\boldsymbol{x}_j^\top \boldsymbol{\theta})^2 < 1 \implies \hat{\boldsymbol{\beta}}_{l_{s-1}+1} = \hat{\boldsymbol{\beta}}_{l_{s-1}+2} = \ldots = \hat{\boldsymbol{\beta}}_{l_s} = 0.$$

The exact calculation of $\max_{\boldsymbol{\theta} \in \mathcal{R}} \sum_{j=l_{s-1}+1}^{l_s} (\boldsymbol{x}_j^\top \boldsymbol{\theta})^2$ is difficult. Hence, we propose to use relaxed methods. For dome region $\mathcal{R} = \{\boldsymbol{\theta} \mid \|\boldsymbol{\theta} - \boldsymbol{\theta}_c\|_2^2 \leq r^2 \wedge \boldsymbol{u}^\top \boldsymbol{\theta} \leq a\}$ ($\|u\|_2 = 1, a < \boldsymbol{u}^\top \boldsymbol{\theta}_c < a + r$), we have

$$\max_{\boldsymbol{\theta} \in \mathcal{R}} \sqrt{\sum_{j=l_{s-1}+1}^{l_s} (\boldsymbol{x}_j^\top \boldsymbol{\theta})^2} = \max_{\boldsymbol{\theta}\|\boldsymbol{\theta}\|_2^2 \leq r^2 - (\boldsymbol{u}^\top \boldsymbol{\theta}_c - a)^2 \wedge \boldsymbol{u}^\top \boldsymbol{\theta} = 0} \|\boldsymbol{X}_s \boldsymbol{\theta}\|_2$$

$$+ \max_{\boldsymbol{u}^\top \boldsymbol{\theta}_c - a \leq t \leq r} \|\boldsymbol{X}_s(\boldsymbol{\theta}_c - t\boldsymbol{u})\|_2$$

$$\leq \|\boldsymbol{X}_s(I - \boldsymbol{u}\boldsymbol{u}^\top)\|_2 \sqrt{r^2 - (\boldsymbol{u}^\top \boldsymbol{\theta}_c - a)^2}$$

$$+ \max(\|X_s(\boldsymbol{\theta}_c - r\boldsymbol{u})\|_2, \|X_l(\boldsymbol{\theta}_c - (\boldsymbol{u}^\top \boldsymbol{\theta}_c - a)\boldsymbol{u})\|_2),$$

where $\boldsymbol{X}_s = (\boldsymbol{x}_{l_{s-1}+1} \ \boldsymbol{x}_{l_{s-1}+2} \ \ldots \ \boldsymbol{x}_{l_s})$. Using Frobenius norm instead of operator norm, we can eliminate some features with little cost.

## D.2  Experiments

First, we compared the number of screened variables among the four dynamic safe screening methods. We solved the group Lasso problem using the subsampled leukemia dataset (dense data with 50 samples and 7128 features) and $\lambda = \frac{1}{100} g^\star(\boldsymbol{X}^\top \boldsymbol{y})$. We used block coordinate ISTA as the iterative algorithm and screened the variables for 50 iterations each. Figure 3a shows the ratio of the unelimated groups at each iteration. As guaranteed theoretically, we can see that Dynamic

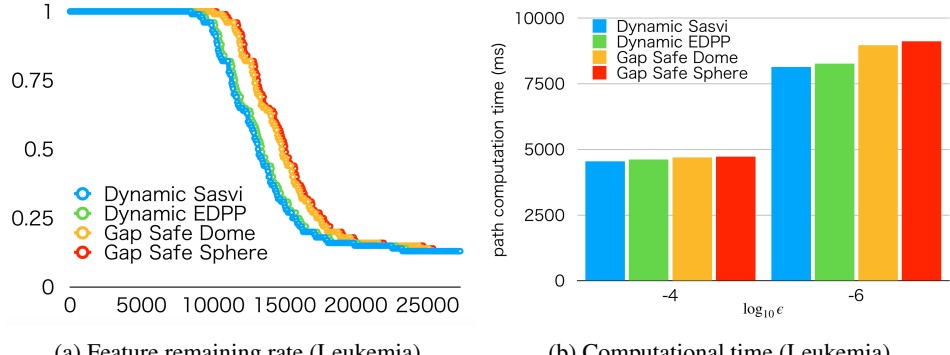

(a) Feature remaining rate (Leukemia).    (b) Computational time (Leukemia).

Figure 3: (a): Feature remaining rate of each iteration for group Lasso on subsampled leukemia (dense data with $n = 50, d = 7128$). (b) Average computational time of the group Lasso path on subsampled leukemia (dense data with $n = 50, d = 7128$).

Sasvi eliminates more variables in earlier steps compared with both Gap Safe Dome and Gap Safe Sphere. The figure also shows that the Dynamic EDPP, which is a relaxed version of Dynamic Sasvi, eliminates almost the same number of feature groups as Dynamic Sasvi.

Next, we compared the computation time of the path of the group Lasso solutions for various values of $\lambda$. We used $\lambda_j = 100^{-\frac{j}{9}} g^\star(\boldsymbol{X}^\top \boldsymbol{y})$ $(j = 0, \ldots, 9)$. The iterative solver stops when the duality gap is smaller than $\epsilon(P(\boldsymbol{0}) - D(\boldsymbol{0}))$. Note that $P(\boldsymbol{0}) - D(\boldsymbol{0})$ makes the stopping criterion independent of the data scale. We used the leukemia dataset. We subsampled the data 50 times and ran all the methods for the same 50 subsamples. For each subsamples, we randomly decided feature groups. Figure 3b shows the average computation times. For all settings, dynamic Sasvi and dynamic EDPP outperform Gap Safe Dome and Gap Safe Sphere.