# OpenReview forum: "Dynamic Sasvi: Strong Safe Screening for Norm-Regularized Least Squares"
_NeurIPS.cc/2021/Conference — NeurIPS 2021 Poster_

### Official Review · Reviewer_grp2 · 2021-07-16

**Rating:** 6
**Confidence:** 5

**Summary:**

The paper proposes a screening test for sparse models, i.e. a way to remove features from the problem once it is sure that they are not active at optimum. The rule is a generalization of the popular Gap Safe screening.
As in GSR, a dual safe region is constructed using strong convexity of the dual objective (obtained from Lipschitz smoothness of the datafit in the primal)

**Main Review:**

The paper proposes to use a Dome region instead of the Gap Safe sphere region, improving upon the dome region that was proposed in the Fercoq et al "Mind the duality gap..." 2015 paper. Note that this does not allow to handle more problems than the Gap Safe framework, but only improves the number of screened variable.
The paper is wwell-written and easy to follow for someone familiar with the literature.


Reading the paper, I felt that the theory was promising, and that it would lead to significant speedups in the computation of Lasso solution. The experimental results somehow did not live up to it: the improvements in timings compared to Gap Safe Screening rule are marginal, as visible on Fig 2. Given that there now exist solvers based on aggressive screening such as Blitz or Celer (Johnson and Guestrin 2016, Massias, Gramfort and Salmon 2018) that outperform Gap Safe Screening rules, I feel the practical results are too low for a venue such as NeurIPS.

Other comments:
The formula L97 is understandable when one is familiar with the littérature but incorrect, as the LHS is a subset of the dual space while the RHS is a subset of the primal space.  (EDIT: I was wrong, thanks for the clarification)

Cosmetic:
In 5 the variable k tends to denote an integer, can you use another variable?
The legend of Fig 1 is very hard to read, as the captions are not separated; the equations are also very small.
in Fig 2 the labels are also impossible to read.
What do you mean by "density of n = 50, p=7129" ?

**Time Spent Reviewing:**

1

---

> ### Author Response · Authors · 2021-08-10
> **Response to your questions**
>
> We thank you for your constructive comments. We will answer your questions.
>
> “Given that there now exist solvers based on aggressive screening such as Blitz or Celer that outperform Gap Safe Screening rules, I feel the practical results are too low for a venue such as NeurIPS.“
>
> Although aggressive screening can only disable some features temporary, safe screening can eliminate some features eternally. Hence, our strong safe screening rules are still effective if we use aggressive screening based solver or working set based solver.
>
> “The formula L97 is ... incorrect, as the LHS is a subset of the dual space while the RHS is a subset of the primal space.“
>
> The left hand side is also a subset of the primal space. Since $X^\top\theta$ is a vector in the primal space, $\partial g^\star(X^\top\theta)$ is a subset of the primal space.
>
> “Cosmetic: ...“
>
> Thanks for your comments about cosmetic. We will edit the paper based on your comments.

---

> > ### Comment · Reviewer_grp2 · 2021-08-18
> > **answer to rebuttal**
> >
> > I thanks the authors for their answer. After reading the other reviews and the rebuttal,  I am still somehow disappointed by the numerical gains, but I think the derivation of the new screening rule is interesting enough and have increased my score.
> >
> > I encourage the authors to release their code for numerical benchmarks with other screening rules and sparse solvers.

---

> > > ### Author Response · Authors · 2021-08-25
> > > **Thank you again for your feedback**
> > >
> > > Thank you again for your feedback. We will release the code at Github and put the information in the main paper.

---

### Official Review · Reviewer_dgBv · 2021-07-16

**Rating:** 7
**Confidence:** 4

**Summary:**

This paper presents a new framework for safe screening based on Fenchel-Rockafellar duality, which generalizes a few existing algorithms in the literature. Given the proposed framework, a tight dome-shape safe region can be constructed for dual variables of Lasso-like problems, so that the zeros in primal solutions can be detected more quickly. The paper has discovered the connection between the new framework and some previously proposed methods, such as Sasvi and EDPP. Theoretically it also shows that the new safe region is always smaller than Gap Safe Dome and Gap Safe Sphere in the literature, which is supported by empirical evidence from experiments.

**Limitations And Societal Impact:**

Yes.

**Main Review:**

Originality: Given existing works in the literature, the new framework for safe screening proposed in this paper does not seem to be super novel, but I am glad that in the first few sections the authors have a quite nice summary on the general ideas for attacking the safe screening problem, and clearly differentiate this work with previous ones.

Quality: The technical quality of this paper is relatively high. Though it looks to me that the most important idea is just to have the upper bound tightened in Theorem 5, proposing its exact form for the Lasso-like problem seems non-trivial, which is more complicated than the constant upper bound proposed previously and may need more careful treatment in terms of math. The result showing its connection to Sasvi is new and interesting. It is also nice to see that the paper theoretically proves that the obtained safe region from such tightening trick is smaller than Gap Safe Dome and Gap Safe Sphere. One thing that is not so clear to me is the application to Elastic-Net in the supplementary material, which is to show the broader applicability of the framework, but technically speaking the function $g$ there is not a norm, which violates Eq. (5). So it the screening method still valid? I am also curious if the result can be extended to group Lasso as well? In the experiments, it would also be ideal to make comparisons on other norms.

Clarity: Overall the paper is well-written. The flow for presenting such (generalized) framework is clear, such that readers can easily tell how it is different from existing algorithms. The figure showing the safe regions for different methods is also helpful for understanding.

Significance: The proposed method, dynamic Sasvi and dynamic EDPP, seem readily applicable to Lasso-like problems to speed up the solving procedure. The framework could be potentially applied to other settings for faster solvers. So this work should be a valuable addition to this research area.


Other detailed comments:

1. In theorem 11, a particular convergent sequence of $\theta^t$ is constructed. For general norms, is there any guidance or recipe that could be provided to help design such sequence?

2. If I understand correctly, the difference between original Sasvi and dynamic Sasvi is just the choice of $\theta^t$ and $\beta^t$ that define the safe region, but I did not see a clear explanation for the latter being faster, so it would be better to clarify a little bit. Also, in general is there any insight on how different choices of $\theta^t$ and $\beta^t$ affect the efficiency of screening? Same for EDPP and dynamic EDPP.

3. In the experiment, dynamic EDPP seems to be faster than dynamic Sasvi, but it looks like the safe region of EDPP is larger than Sasvi theoretically. So the result is a bit counterintuitive. Is the reason that some computation is faster for EDPP region than Sasvi? Can authors give a comment on this?


**Time Spent Reviewing:**

5

---

> ### Author Response · Authors · 2021-08-10
> **Response to your questions**
>
> We thank you for your constructive comments. We will answer your questions.
>
> “One thing that is not so clear to me is the application to Elastic-Net ... which violates Eq. (5).“
>
> In appendix C, we use only Thorem 5. Since Theorem 5 is not based on Eq. (5), the discussion in appendix C is valid. Note that the rules in section 4 are not applicable to Elastic-Net.
>
> “... if the result can be extended to group Lasso as well?“
>
> Yes. Since group Lasso satisfies Eq. (4) and Eq. (5), group Lasso is Lasso-like. We can use dynamic Sasvi and dynamic EDPP for it.
>
> For other detailed comments
> 1. For genral norms, how can we construct a convergent sequence of $\theta^t$?
>
> If the computation of the dual norm of $g$ costs not so large, we can construct a convergent sequence of $\theta^t$ by replacing the infinity norm in Theorem 11 with the dual norm. If the computational costs of the dual norm of $g$ is large, an efficient method is unknown. This is a potential future work. We will discuss this further either in the main paper or in the supplementary material.
>
> 2. Why dynamic Sasvi is faster than original Sasvi? How different choices of $\theta^t$ and $\beta^t$ affect the efficiency of screening?
>
> As stated in Theorem 10, the squared radius of dynamic EDPP is smaller than the duality gap. Hence, the regions of dynamic Sasvi and dynamic EDPP are narrow when $\theta^t$ and $\beta^t$ are better vectors. Although original Sasvi can use the solution to the problem with another regularization parameter, our dynamic Sasvi can work with better feasible vectors in later steps of optimization. Hence, dynamic Sasvi is faster than original Sasvi. This is the same for EDPP and dynamic EDPP.
>
> 3. Why dynamic EDPP is faster than dynamic Sasvi?
>
> The sphere region of dynamic EDPP is a little simpler than the dome region of dynamic Sasvi. This makes dynamic EDPP a little faster. Note that dynamic Sasvi always removes at least the same number of features to dynamic EDPP.

---

> > ### Comment · Reviewer_dgBv · 2021-08-30
> > **Post-rebuttal comments**
> >
> > After reading the author response and other reviews, I would like to keep my original score.

---

### Official Review · Reviewer_Cu9Q · 2021-07-18

**Rating:** 6
**Confidence:** 3

**Summary:**

 The concern of  the paper is the design of screening rule for sparsity-based regularized convex  optimization problem  such as the Lasso-type problem.  The aim is to screen out  irrelevant parameters during  the optimization  procedure. The  designed screening rule hinges on Fenchel-Rockafellar duality and  encompasses existing  methods (Gap Safe Sphere, Gap Safe Dome, Sasvi). The  paper establishes that the  constructed  safe region for the dual solution is of reduced size than the  ones the aforementioned methods. Hence, the new rule induces a more efficient screening. Empirically evaluation on Lasso problem show that the proposed screening rule brings computation gain when compared to existing methods.

**Ethical Concerns:**

The  paper does not raise  any ethical issues.


**Limitations And Societal Impact:**

Unless I  was mistaken, the paper does not discuss the broader impact of the proposed approaches. On my view the work does not raise specific issues


**Main Review:**

- Overall, the paper is well written and well  organized. The rationale behind the new screening rule is justified and the  novelty brought  by  the approach is stated clearly.  The key element  to  the rule  is that the dual of  the problem  may be upper- bounded by a term depending on the conjugate of the fitting term provided a primal/dual inequality is met.  In  the context of the Lasso-type problems, the upper bound exploits convexity hence,  it appears somehow obvious.

- The safe region of the advocated screening rule is expressed as the intersection  of  a sphere and  a half space. Compared to Gap Safe Dome or Gap Safe Sphere, that  safe region is of reduced size. Hence, the  obtained screening rule is theoretically shown to be stronger. This fact is illustrated by  empirical findings. Overall the algorithmic derivation and empirical evaluation represents a novelty of the work. The empirical findings show  substantial computation gain that is of interest to practitioners. Notice that a side result of  the paper is  Dynamic EDPP, a relaxation of the proposed screening rule that proves slightly faster  while keeping interesting screening abilities.

- Theorem 5 is trivial provided the safe region expression Lines 116-117.

- The overall safe region construction is devised for a general Lasso-like  problem. However, the remainder of the paper  is restricted to the sole Lasso. Though the empirical findings for Lasso aree convincing, it will be interesting to show the results for other regularizers (for instance elastic-net,  group lasso to name a few).
-When used to compute the Lasso regularization path, Dynamic EDPP provides slightly better computation time than Dynamic Sasvi. Is this due to a simplified  screening rule?

- Another  question raised is how the proposed Fenchel Duality based derivations might be lifted to problems such as spare logistic regression learning? Gap Sphere method does so. At least the paper should discuss this point.

- Other comments
The caption or Figure 1 has to reformatted.


**Time Spent Reviewing:**

8

---

> ### Author Response · Authors · 2021-08-10
> **Response to your questions**
>
> We thank you for your constructive comments. We will answer your questions.
>
> “..., Dynamic EDPP provides slightly better computation time than Dynamic Sasvi. Is this due to a simplified screening rule?“
>
> Yes. The number of eliminated features of Dynamic EDPP never be more than that of Dynamic Sasvi. Dynamic EDPP outperforms Dynamic Sasvi only in the cost of screening.
>
> “... how the proposed Fenchel Duality based derivations might be lifted to problems such as sparse logistic regression learning?“
>
> Application to other problems such as logistic regression learning are discussed in appendix C. By our framework, we can derive smaller safe region than gap safe sphere.
>
> “The caption for Figure 1 has to reformatted.“
>
> Thanks for your comments. We will reformat it.

---

> ### Comment · Reviewer_Cu9Q · 2021-08-23
> **answer to rebuttal**
>
> I thank the authors for providing responses to the raised points. After reading the other reviews and the rebuttal, it seems that some points are partly covered, especially the empirical evaluations of the method on other Lasso-like problems (Elastic Net, Group-Lasso). Nevertheless the theoretical derivation of the screening rule and the fact that its safe region is theoretically smaller than Gap Safe Dome and Gap Safe Sphere are interesting.

---

> ### Author Response · Authors · 2021-08-25
> **Thank you again for your feedback**
>
> Thank you again for your feedback. We are glad of your evaluation for our theoretical results. We will add some results on other Lasso-like problems.

---

### Decision · Program_Chairs · 2021-09-27

**Decision:**

Accept (Poster)

**Comment:**

All reviewers agrees that the derivation of a new screening rule is interesting, and that the mathematical
results are strong and leads to (slight) computational acceleration compared to SOTA methods.

The authors are encouraged to release their code for numerical benchmarks with other screening rules and sparse solvers.
(there exists a open-source project about that, named BenchOpt)